# Understanding Parental Perceptions of Content-Specific Barriers to Preventing Unintentional Injuries in the Home

**DOI:** 10.3390/children10010041

**Published:** 2022-12-25

**Authors:** Mikiko Oono, Yoshifumi Nishida, Koji Kitamura, Tatsuhiro Yamanaka

**Affiliations:** 1Artificial Intelligence Research Center, National Institute of Advanced Industrial Science and Technology, Koto-ku, Tokyo 135-0064, Japan; 2Department of Mechanical Engineering, School of Engineering, Tokyo Institute of Technology, Meguro-ku, Tokyo 152-8552, Japan; 3Ryokuen Children’s Clinic, Yokohama-shi 245-0002, Japan

**Keywords:** home injury, content-specific barriers, 3Es of injury prevention, children

## Abstract

Background: Preventable injuries are the leading cause of death in children around the world, including in Japan. As children under the age of 5 years spend most of their time at home, home injury prevention is critical for child safety. The purpose of this study was to identify specific, focused, and precise barriers against injury prevention practice. Methods: We conducted an online survey to examine the barriers faced by parents when taking actions to prevent home injuries. Results: The results revealed common reasons why parents do not or cannot take a recommended action across injury types, and that the magnitude of importance for a specific barrier depends on the type of injury. Conclusions: Identifying content-specific barriers could help researchers and educators understand parents’ needs, discuss what barriers are more important than others by injury type, and develop effective strategies based on the 3Es of injury prevention (enforcement, engineering, and education).

## 1. Introduction

Preventable injuries are the leading cause of death in children around the world. Globally, approximately 2000 children under the age of 14 years die as a result of injuries every day [1]. Unintentional injuries were responsible for 1.77 million deaths and 104 million disability-adjusted life-years (DALYs) in 2019. Childhood DALYs accounts for a significant portion of all burdens [2]. Awareness of the importance of injury prevention has been increasing among Japanese people, but preventable injuries remain a leading cause of death in children [3]. Although national statistics are not available, 13,582 children aged 0–12 years in Tokyo, Japan alone were transported to an emergency room (ER) due to preventable injuries during 2019, which means that 37 children per day had to be taken to the ER because of injury [4]. About 68% of these injuries, 80% of which involve falls, suffocations, collision, burns, and pinches, occur in children under the age of 5 years [4]. As children under the age of 5 years spend most of their time at home, home injury prevention is critical for child safety.

Where and how children get injured at home is already commonly known, and information on how to prevent injuries is widely available. However, unacceptable numbers of children still experience unintentional injuries, some of which even lead to death. Regarding home injury, parents are the key to injury prevention, and as such, must take specific actions to create a safe home environment. For instance, using stair gates and window guards, keeping medication out of children’s sight and reach, and anchoring TVs. Some actions are simple, whereas others are complex. Kendrick et al. [5] examined whether home safety interventions with or without the provision of low-cost, discounted, or free equipment reduced injuries and promoted safe behaviors and practices. Although further research is needed, they concluded that home safety interventions produce positive impacts on eight types of safety practices, such as a safe hot tap water temperature, a fire escape plan, and socket covers on unused sockets. However, few studies have been conducted to identify facilitators and barriers that promote behavior change in parents with a focus on childhood home injury. Smithson et al. [6] conducted a systematic review to identify factors affecting the success of interventions at the legal, environmental, and individual levels. In terms of barriers at the individual level, they highlighted the fact that parents’ preventive actions were strongly related to their cultural and socioeconomic backgrounds, age, and experience. Ablewhite et al. [7] conducted a qualitative study and identified five main themes for barriers, including a lack of anticipation with regard to injury risk, a belief that injuries were inevitable, maternal fatigue, and difficulties in adapting the home. A study by Gielen et al. [8] revealed that family income, poor housing quality, and environmental barriers were associated with injury prevention practice.

The purpose of the present study was to identify more specific, focused, and precise barriers against injury prevention practice. The barriers identified in previous studies were for home injury in general, and no research has been conducted to identify specific barriers according to a particular home injury. The underlying problem is that parents are required to take particular actions that are already determined, and the barriers parents face or the burden they feel must be very different. We believe that gaining a better understanding of parental perceptions of content-specific barriers could create an innovative pathway for parent education for injury prevention.

## 2. Methods

We conducted an online survey to examine the barriers faced by parents when taking actions to prevent childhood home injuries. We recruited parents aged 18–60 years who had at least one child aged between 6 months and 6 years (preschooler) at the time of the study through Cross Marketing Group, Inc., a leading research marketing company in Japan with approximately 3.77 million panels. The selected panels received the survey URL and answered the questions. To develop the survey questions, we selected 11 types of home injuries that are important for Japanese parents from two perspectives: the frequency of injury and the fact that the injury was featured by the Consumer Affairs Agency. Then, we determined the recommended action by each injury type and asked five individuals who had a child to list as many barriers as possible that parents in general might have. After collecting all the lists, we merged the common barriers and finalized the survey. The online survey was conducted on 26–28 September 2022. All analyses were carried out using Microsoft Excel. In this study, IRB review was waived by our research organization because the research involved no more than minimal risk to the online survey respondents (#2022-1254). The respondents agreed to answer the questions after being asked if their age was between 18 and 60 years and if they had a preschooler. They were also informed that they could withdraw from the survey at any time. Four examples of safety goods are shown in Figure 1.

## 3. Results

As planned, 600 parents (187 males and 413 females) participated in this study. The mean age of the respondents was 35.16 years (range, 19–59 years). The survey results are shown in Table 1.

When asked whether they were concerned that their child would be injured, 33% of the respondents said “strongly agree”, 49% “agree”, 14% “somewhat disagree”, and 4% “disagree”. In this section, we focus on the response of “Other” for each question to understand the specific reasons regarding why the respondents do not take the recommended preventive action. Table 1 shows the specific numbers and percentages for each question.

Regarding the use of stair gates, about 40% (164/427) of the respondents who did not use stair gates said that they lived in a home without stairs. Among the respondents who lived in a home with stairs and selected “Other”, some of the responses were “Sometimes it’s necessary to learn about injury risks”, “I can’t install the gates because of the design of the stairs”, “I worried that the gates would come off if my child grabbed them”, and “My child is old enough to take care of him/herself”. Among the respondents who selected “Other” for the use of child-proof locks on doors or window stops to prevent falls from a window or balcony, some noted, “I teach my child about the risk”, “I don’t let my child play near the balcony”, “I’ve never thought about using them”, and “I plan on using them”. When the respondents were asked if they let their child sleep in an adult bed, the responses for “Others” included “My child sleeps between the parents”, “My child cries if we don’t sleep together”, “It’s difficult to breastfeed the child if he/she doesn’t sleep in the bed”, and “My child sleeps on a futon on the floor”. Regarding the use of corner guards, examples of responses for “Other” included “I’m worried my child will eat part of the corner guards”, “My home doesn’t have any sharp edges”, “I haven’t really cared about it before”, and “I think it’s overprotection”. When asked about keeping water in the bathtub, seven respondents selected “Other”. A few respondents said that they kept water in the bathtub for no reason, and one respondent said that he/she kept water in the bathtub because they used it for bathing the next day. Regarding the use of finger pinch guards, the responses for “Other” included “I did not know there were finger pinch guards” (16/51), “I always tell my child not to touch door hinges”, and “My child is wary about opening doors”. When asked about the use of a steamless rice cooker, about 60% of the respondents who did not use them said, “I didn’t know there were steamless rice cookers”. Among those who selected “Other” for this question, the responses included “I don’t use a rice cooker when my child is awake”, “I don’t have a rice cooker”, and “I don’t let my child in the kitchen”. Among the respondents who did not use a steamless rice cooker, 20% said that they did not keep their current rice cooker out of the reach of children. Some respondents said that there were no places to keep it out of the reach of children, the only place they could put it was on a low shelf in the first place, and they could not keep it because of the location of electric outlets. When asked about the use of an electric kettle with a leak prevention function, 81 respondents selected “Other”. Sixty-eight of them said that they did not own or use a kettle, but some said, “I think it’s overprotection”, “I don’t let my child in the kitchen”, and “I use a water dispenser”. When asked if they always cut cherry tomatoes or grapes into four, answers for “Other” included “I do not give them to my child” (30/100), “I cut them into two” (19/100), and “My child does not like them” (18/100). Regarding whether they kept button cells (batteries) out of the reach of their child, most respondents (88%) responded that they did. Among those who selected “Other”, the responses included “I do not have or use button cells at home” (14/16), “I change button cells with my child” (1/16), and “I’ve never thought about it”. Lastly, when asked about whether they anchored their TV to prevent it from tipping over, the reasons not for anchoring the TV included “I don’t know which one to buy”, “I tell my child not to touch the TV”, and “There is enough space in front of TV in case the TV tips over”.

Based on the survey responses, the authors identified the following five common themes that hindered parents’ safety practices: (1) beliefs about child supervision, (2) a hassle, (3) expectations about children’s behavior, (4) assumptions about injury severity, and (5) the lack of information regarding safety goods.

**Theme 1.** The respondents were not taking the recommended preventive actions because they believed that they could prevent their children from injury through close supervision. This parental belief was the top reason given in the case of falls on the stairs (*always uses the stairs with an adult*), falls from the balcony (*do not let my child go onto the balcony alone*), and falls from an adult bed (*always stay close*). This belief was also observed with regard to drowning in the bathtub (*always takes a bath with adults*), scalds from a rice cooker (*always watch my child*), and scalds from an electric kettle (*always watch my child*).

**Theme 2.** The respondents were not taking the recommended preventive actions because they considered such actions to be a hassle. One of the characteristics of preventive actions for home injuries is that parents are required to use safety goods or devices. Answers of “*It’s a hassle to…*” were observed in many cases investigated in this study.

**Theme 3.** The respondents were not taking the recommended preventive actions because they did not think that their child behaved in a way that would lead to injury. For instance, some respondents said, “*I don’t think my child swallows them whole (tomatoes/grapes)*” and “*I don’t think my child ever climbs up to the window*”. In addition, some respondents did not think that their child did anything dangerous because they educated their child about injury risks. For example, some said, “*I tell my child to chew them (tomatoes/grapes)*”, “*I teach my child about the risk*”, and “*I always tell my child not to touch door hinges*”.

**Theme 4.** The respondents were not taking the recommended preventive actions because they did not think their child would experience a severe injury. For instance, some examples for this parental assumption were, “*I don’t think that my child can get severely burnt by steam*”, “*I don’t think that my child would be seriously injured if the TV tipped over*”, “*I think it’ll be fine even if my child swallows a button cell*”, and “*If any food got stuck in my child’s esophagus, I think could remove it*”.

**Theme 5.** The respondents were not taking the recommended preventive actions because of a lack of information on safety goods. For instance, some respondents said, “*I’m not sure which ones are good to buy*”, “*I don’t know where to buy them*”, “*I didn’t know electric kettles with such a function were available*”, and “*I didn’t know there were steamless rice cookers*”. Thus, improving the accessibility of information could help reduce the hassle experienced by parents.

## 4. Discussion

In this study, we attempted to clarify parental perceptions of content-specific barriers to preventing unintentional injuries in the home. Although more than 80% of the respondents said they were concerned about keeping their children safe from unintentional injury, many were not taking the recommended preventive actions because of a variety of barriers (see Table 2).

The results of this study revealed that many parents rely heavily on their close supervision to protect their child from injury (Theme 1). Although child supervision is important, evidence of its effectiveness to prevent injury remains limited [6]. We also found that many parents do not take the recommended actions because they consider such actions to be a hassle (Theme 2). Most of respondents care about unintentional injury, but they put off taking necessary actions because of the hassle involved. Regarding the process of completing a preventive action among parents, at least four steps were identified: (1) select, (2) buy, (3) install, and (4) use. These four steps were strongly related to whether the action was a hassle. Thus, a new system needs to be developed to reduce such barriers. For instance, creating a list of effective safety products with information about how they can be acquired would be helpful. As most people now use a smartphone and online shopping services, the first two steps could be completed in a matter of seconds if such a list were available. Regarding steamless rice cookers and electric kettles with added safety features, about 60% and 40% of the respondents, respectively, had not even reached the initial *Select* phase because they did not know the product existed. Regarding parental expectations and assumptions (Themes 3 and 4), parental education plays an important role in enhancing parental perceptions of susceptibility and injury severity. Furthermore, to improve the accessibility of information (Theme 5), conducting a social marketing campaign in addition to parental education could promote the use of safety goods and devices.

This study reports four important findings. First, product improvement plays a significant role in diminishing barriers for parents. For instance, the main reason for not using corner guards was that the child always removed them, and one respondent was even worried that his/her child would eat them. In addition, some of the respondents hesitated to use stair gates, corner guards, or finger pinch guards because they would not be able to be removed without leaving residue. Surely, these issues can be solved through product modifications. Second, even if the reasons are the same, the magnitude to which the reasons hinder parents’ preventive actions varies according to injury type. In other words, each barrier has its own weight, and the weight of that barrier changes based on the type of injury. For instance, while hassle is the top reason why the respondents do not take action to prevent stair injuries, the perceived difficulty of installing anchors to prevent TVs from tipping over was a more serious problem than the hassle to install them. We cannot simply compare the magnitude of importance of a specific reason, such as the hassle of installation, between two injury types; however, the results of this study clearly showed that a specific reason can become a serious barrier in some injury cases, and is less important in others. Understanding the specific reason in detail by injury type is necessary to prioritize which barriers to mitigate first. Third, after identifying specific barriers that need to be worked on, it would be useful to develop possible strategies based on the 3Es (enforcement, engineering, and education) of injury prevention [9]. Table 3 shows a summary of possible strategies based on the 3Es. As mentioned previously, product modifications or improvements play a huge role in changing the current situation. The development of safety goods or devices listed in Table 3 could be expected to have a great impact on society. Some respondents said that the cost of safety goods is a barrier against taking action. Therefore, cost control for safety goods or the distribution of coupons could help promote preventive practices. Fourth, understanding parental perceptions of content-specific barriers is critical to strengthen *systems* to have a population-level impact. From the viewpoint of the key features of “a system approach” described by Taylor et al. [10], we can consider how alternative strategies could be embedded to help people overcome specific barriers within systems, with whom we should build partnerships, and what data should be collected for program evaluations based on identified content-specific barriers.

This study had some limitations. First, even though we developed the survey questions with five parents, we may not have devised enough response choices for the respondents. For the question about stair gates, we did not ask whether the respondents used them at either the top or bottom of the stairs. The best practice is to install hardware-mounted gates at both ends, but some respondents did not provide a complete response about barriers because they thought they used the gates as recommended. Second, we did not ask the respondents about the age of their child, so we could not analyze the characteristics of parental perceptions by child age. If the respondent had a child aged only 6 months, and that child was the first for him/her, then they did not have much experience with childhood injuries and thus might not have felt it necessary to take preventive actions. For the respondents who had a 5- or 6-year-old child, the important period for preventing drowning in the bathtub drowning was almost over. Therefore, some of these respondents were keeping water in the bathtub for disaster preparedness or washing clothes. Third, we did not ask the respondents about individual characteristics such as sex, education, socioeconomic status, or housing situation (rental or ownership). Different barriers would be faced by different groups of parents, and thus, further research is needed to explore more effective preventive strategies. Despite these limitations, the results of the present study clearly demonstrate the importance of understanding parental perceptions of content-specific barriers, which could help researchers and educators more effectively (1) understand parents’ needs, (2) discuss what barriers are more important than others by injury type, and (3) develop effective strategies based on the 3Es.

## 5. Conclusions

Preventable injuries at home are a serious health problem for children around the world and have become a crucial theme in the United Nations Sustainable Development Goals [11]. To change the current situation, we investigated parental perceptions of content-specific barriers for 11 injury types. The results indicated that there were common reasons why parents do not or cannot take a recommended action across injury types, and that the magnitude of importance for a specific barrier depends on the type of injury. In conclusion, we strongly believe that exploring parental perceptions of content-specific barriers could have an important impact on moving the field of injury prevention forward.

## Figures and Tables

**Figure 1 children-10-00041-f001:**
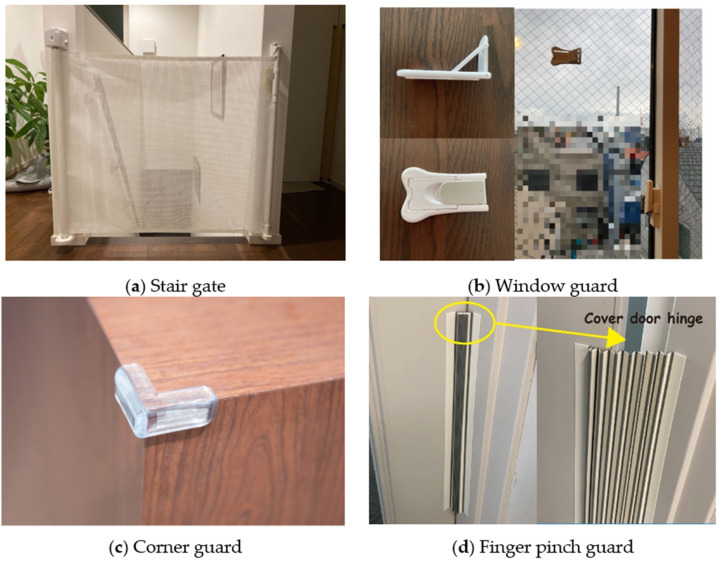
Examples of safety goods: (**a**) Stair gate at the top of the stairs. Stair gates need to be used at both the top and bottom of the stairs; (**b**) Window guard. Window guards limit the distance a window can be opened; (**c**) Corner guard; (**d**) Finger pinch guard.

**Table 1 children-10-00041-t001:** Survey results.

	Question	n	%
1	Please indicate your level of agreement with the following statement: I’m very concerned about keeping my child safe from unintentional injury.		
	□ Strongly agree	201	33.5
	□ Agree	291	48.5
	□ Somewhat disagree	83	13.8
	□ Disagree	25	4.2
2	Do you use safety gates at the top and bottom of the stairs?		
	⯎ Yes	173	28.8
	⯎ No	427	71.2
	If yes, are there times when the stair gates are not closed?		
	⯎ Yes	41	23.7
	⯎ No	132	76.3
	If no, please select all that apply:		
	□ They’re a hassle to install.	35	8.2
	□ It’s a hassle to select the appropriate gates.	11	2.6
	□ My older child leaves the gates open.	20	4.7
	□ My child is safe because he/she always uses the stairs with an adult.	59	13.8
	□ I can’t drill holes to install the gate properly.	29	6.8
	□ I’m worried that I may not be able to remove them when they’re no longer needed.	24	5.6
	□ I’m worried that I may not be able to sell them in a secondhand shop when they’re no longer needed.	2	0.5
	□ I’m doubtful about the effectiveness of the gates.	28	6.6
	□ I don’t want to damage my home.	32	7.5
	□ Stair gates are expensive.	27	6.3
	□ Other (please specify)	227	53.2
3	Have you installed child-proof locks on door and/or window stops to prevent falls from a window or balcony?		
	⯎ Yes	199	33.2
	⯎ No	401	66.8
	If yes, are there times when an adult has left them unlocked?		
	⯎ Yes	43	21.6
	⯎ No	156	78.4
	If no, please select all that apply:		
	□ I’m not sure which ones are good to buy.	50	12.5
	□ I don’t know where to buy them.	33	8.2
	□ They’re a hassle to install.	52	13.0
	□ I (or other family members) may leave them unlocked after installation.	51	12.7
	□ I (or other family members) may feel like using them is a hassle.	39	9.7
	□ They aren’t very aesthetically appealing.	6	1.5
	□ I don’t think my child ever climbs up to the window.	37	9.2
	□ I think my balcony railings are high enough to prevent falls.	45	11.2
	□ I don’t let my child go onto the balcony alone.	131	32.7
	□ Other (please specify)	71	17.7
4	Do you let your child sleep in an adult bed?		
	⯎ Yes	280	46.7
	⯎ No	320	53.3
	If yes, please select all that apply:		
	□ There’s no other place for my child to sleep.	84	30.0
	□ I sometimes think that it’ll be fine for just one day.	16	5.7
	□ My child won’t fall because I use a bed guard.	38	13.6
	□ My child won’t fall because I always stay close.	123	43.9
	□ My child won’t fall because I let him/her sleep on the side of the wall.	71	25.4
	□ I place something soft such as a stuffed toy, mat, or bedding around the bed in case my child falls.	73	26.1
	□ Other (please specify)	22	7.9
5	Do you use corner guards?		
	⯎ Yes	225	37.5
	⯎ No	375	62.5
	If no, please select all that apply:		
	□ They make the furniture look bad.	38	10.1
	□ My child always removes them.	141	37.6
	□ I’m worried that I may not be able to remove them when they’re no longer needed.	57	15.2
	□ Severe injuries rarely occur.	81	21.6
	□ Other (please specify)	83	22.1
6	Do you keep the water in the bathtub even after bath time for your child is over?		
	⯎ Yes	368	61.3
	⯎ No	232	38.7
	If yes, please select all that apply:		
	□ I keep water in the bathtub in case of a disaster.	57	24.6
	□ I can’t tell my spouse or parent-in-law to drain the water from the bathtub.	7	3.0
	□ Bathing times vary among family members.	73	31.5
	□ I want to use the water for cleaning and washing clothes.	87	37.5
	□ I don’t let my child go in the bathroom alone.	20	8.6
	□ My child always takes a bath with adults.	47	20.3
	□ Other (please specify)	7	3.0
7	Do you use finger pinch guards?		
	⯎ Yes	100	16.7
	⯎ No	500	83.3
	If no, please select all that apply:		
	□ It is a hassle to use them.	90	18.0
	□ I’m not sure which ones are good to buy.	148	29.6
	□ They are expensive.	26	5.2
	□ I’m worried that I may not be able to remove them when they’re no longer needed.	31	6.2
	□ I don’t think it’s necessary because I use a door closer.	48	9.6
	□ I don’t think there are spaces or gaps large enough to catch the fingers.	169	33.8
	□ Other (please specify)	51	10.2
8	Do you use a steamless rice cooker?		
	⯎ Yes	75	12.5
	⯎ No	525	87.5
	If no, please select all that apply:		
	□ Steamless rice cookers are expensive.	76	14.5
	□ I didn’t know there were steamless rice cookers.	311	59.2
	□ I don’t want to buy one because the one I use now still works fine.	122	23.2
	□ I don’t think it’s necessary to have one because I keep my rice cooker out of the reach of children.	81	15.4
	□ I don’t think my child can get burnt because I always watch him/her.	35	6.7
	□ If my child starts to place his/her hand over the rice cooker, I stop him/her.	16	3.0
	□ I don’t think that my child can get severely burnt by steam.	15	2.9
	□ Other (please specify)	19	3.6
	If you do not use a steamless rice cooker, do you keep your current rice cooker out of the reach of children?		
	⯎ Yes	420	80.0
	⯎ No	105	20.0
9	Do you use an electric kettle with a leak prevention function?		
	⯎ Yes	136	22.7
	⯎ No	464	77.3
	If no, please select all that apply:		
	□ Electric kettles with a leak prevention function are expensive.	31	6.7
	□ I didn’t know electric kettles with such a function were available.	188	40.5
	□ I don’t think it’s necessary to have one because I keep my kettle out of the reach of children.	120	25.9
	□ I don’t think my child can get scalded because I always watch him/her.	23	5.0
	□ I don’t want to buy one because the one I use now still works fine.	52	11.2
	□ If my child starts to touch the kettle, I stop him/her.	30	6.5
	□ Other (please specify)	81	17.5
10	Do always you cut cherry tomatoes or grapes into four?		
	⯎ Yes	258	43.0
	⯎ No	342	57.0
	If no, please select all that apply:		
	□ It’s a hassle to cut cherry tomatoes or grapes into four.	79	23.1
	□ I sometimes think that it’ll be fine for just one day.	28	8.2
	□ I don’t think my child swallows them whole.	70	20.5
	□ I tell my child to chew them.	72	21.1
	□ My child would only suffocate if he/she were unlucky.	18	5.3
	□ If any food got stuck in my child’s esophagus, I think I could remove it.	9	2.6
	□ I don’t want to waste the juice by cutting them.	21	6.1
	□ Other (please specify)	100	29.2
11	Do you keep button cells (batteries) out of your child’s reach?		
	⯎ Yes	527	87.8
	⯎ No	73	12.2
	If no, please select all that apply:		
	□ I’m not sure where button cells are used in my house.	21	28.8
	□ Older siblings move button cells using a chair or other things for climbing even if I keep them out of reach.	12	16.4
	□ There are many places in my home that is within the reach of child.	17	23.3
	□ I think it’ll be fine even if my child swallows a button cell.	5	6.8
	□ I don’t think it’s likely that my child would swallow a button cell.	8	11.0
	□ Other (please specify)	16	21.9
12	Do you anchor your TV?		
	⯎ Yes	319	53.2
	⯎ No	281	46.8
	If no, please select all that apply:		
	□ Anchors for the TV are expensive.	20	7.1
	□ They’re difficult to install without help.	67	23.8
	□ They’re a hassle to install.	58	20.6
	□ After comparing the frequency of falls versus the labor and installation costs, I decided not to install them.	29	10.3
	□ I don’t think that the TV will tip over.	64	22.8
	□ I don’t think that my child would be seriously injured if the TV tipped over.	28	10.0
	□ Other (please specify)	46	16.4

**Table 2 children-10-00041-t002:** Percentage of respondents who were not taking the recommended action by injury type.

Type of Injury	Preventive Action Recommended	% of Those Who Were Not Taking the Recommended Action
Fall on stairs	Use stair gates	47.2 *
Fall from a balcony	Use child-proof locks	66.8
Fall from an adult bed	No sleeping in an adult bed	46.7
Sharp corner injury	Use corner guards	62.5
Drowning	Drain water after use	61.3
Finger pinches	Use finger pinch guards	83.3
Rice cooker scald	Use a steamless rice cooker	87.5
Boiling water scald	Use an electric kettle with a leak prevention function	77.3
Choking on tomatoes	Cut them into four	57.0
Button cell ingestion	Keep them out of the reach of children	12.2
TV tip-over	Anchor the TV	46.8

* Respondents who did not have stairs in the home (n = 164) were excluded and the percentage was recalculated.

**Table 3 children-10-00041-t003:** Summary of possible strategies based on the survey results.

Enforcement	Engineer	Education
Change rental agreements (renters allowed to make holes and scratches for installation) Mandate that manufacturers sell TVs with tip-over prevention goods Cost control for safety goods Distribute coupons for safety goods	Develop gadgets that can be removed easily without residue Develop corner guards that cannot be peeled off easily Develop stylish safety devices that parents will want to use Develop automated stair gates that close after use Create new online services for purchasing safety goods	Promote safety goods Teach about the effectiveness of child supervision Create a new education class for older siblings

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
