# Peer review of "Understanding Parental Perceptions of Content-Specific Barriers to Preventing Unintentional Injuries in the Home"

_children, 2022, doi:10.3390/children10010041_

Round 1

Reviewer 1 Report

Thank you for the opportunity to review this analysis of parents perceptions around barriers to not using effective interventions around the home for their under 5's.

I believe that there are a number of places where the paper could be strengthened.

1. The Global Health Estimates or Global Burden of Disease estimates could be used to quote injury death/DALY rates in under 5's in the absence of national data (although there must be national fatal data). 

2. Denise Kendrick and her colleagues in the UK have done extensive research in this area including the publication of systematic reviews (on injury types) and overall interventions. I am surprised that they authors have not quoted this work nor drawn on their extensive recommendations.

3. The authors do discuss the limitation of only using 5 parents to develop the survey questions but they dont explain well enough how the 11 types of injuries were agreed - did they use a Delphi process, with which stakeholders? Results categorised as "other" - particuarly for stairgates (227) need more explanation

4. The table is very comprehensive. However, it might be useful for the reader to understand some of the interventions. For example, what are finger pinch guards? What was coded if parents only had one stairgate at the top or bottom of the stairs and not top AND bottom as the survey question asks?  

5. The results are pretty much a repeat of what is in the table. It would have been more useful to have done some content analysis here in order to come up with themes - which would then support the statements in the recommendations around the 3 important findings.

6. Results are introduced in the conclusion. Would these not be better placed in the results section and then referred back to in the conclusion.

7. The authors could consider additional analysis - for example were there differences between what mothers or fathers thought were the barriers? It would have been useful to also look at the child's age in this regard, but i note that this is identified as a weakness. 

8. Most injury programmes are moving away from 3E's to a systems approach.  The authors could consider discussing recommendations under this more holistic approach. 

Author Response

Dear Reviewer,

Thank you for reviewing my research paper. Please see the attachment.

Reviewer 2 Report

Introduction: Could be improved by adding global data on childhood injuries.

Methods: Please explain/add why only these 11 injuries were included in the survey. Any reason why poisoning was not included? 

Results: Table 1 and related text is repetitive. 

Discussion: The study does not discuss limitations in detail. Family income or socioeconomic status, housing (rental or ownership), parental education may influence child injury prevention practices. These are not collected or mentioned. The survey includes only those who will have access to internet. For some respondents, expense was a concern to buy safety gadgets but is not discussed.   

Author Response

Dear reviewer,

Thank you for reviewing my paper. Please see the attachment.

Mikiko
